# Increased O-GlcNAcylation by Upregulation of Mitochondrial O-GlcNAc Transferase (mOGT) Inhibits the Activity of Respiratory Chain Complexes and Controls Cellular Bioenergetics

**DOI:** 10.3390/cancers16051048

**Published:** 2024-03-05

**Authors:** Paweł Jóźwiak, Joanna Oracz, Angela Dziedzic, Rafał Szelenberger, Dorota Żyżelewicz, Michał Bijak, Anna Krześlak

**Affiliations:** 1Department of Cytobiochemistry, Faculty of Biology and Environmental Protection, University of Lodz, 90-236 Lodz, Poland; anna.krzeslak@biol.uni.lodz.pl; 2Institute of Food Technology and Analysis, Faculty of Biotechnology and Food Sciences, Lodz University of Technology, 90-924 Lodz, Poland; joanna.oracz@p.lodz.pl (J.O.); dorota.zyzelewicz@p.lodz.pl (D.Ż.); 3Department of General Biochemistry, Institute of Biochemistry, Faculty of Biology and Environmental Protection, University of Lodz, 90-236 Lodz, Poland; angela.dziedzic@biol.uni.lodz.pl; 4Biohazard Prevention Centre, Faculty of Biology and Environmental Protection, University of Lodz, 90-236 Lodz, Poland; rafal.szelenberger@biol.uni.lodz.pl (R.S.); michal.bijak@biol.uni.lodz.pl (M.B.)

**Keywords:** mOGT, mitochondria, electron transport chain, O-GlcNAc, VDAC, breast cells

## Abstract

**Simple Summary:**

O-GlcNAcylation is a rapid and reversible posttranslational modification involved in the adaptation of cells to nutrient availability. The interplay of O-GlcNAcylation and phosphorylation controls signal transduction pathways to maintain cellular homeostasis. To date, studies have concentrated on the O-GlcNAcylation of proteins by the nuclear-cytoplasmic isoform of O-GclNAc transferase (ncOGT) whereas the function of mitochondrial OGT (mOGT) is poorly understood. The goal of our research was to investigate how mOGT affects the phosphorylation of mitochondrial proteins and the synthesis of ATP. Based on our results, we suggested that the deregulation of mOGT might be sufficient for changes in electron transport chain activity and ATP synthesis. Inhibition of the energy metabolism of cancer cells is important for potential cancer therapies; therefore, mOGT could be considered as a therapeutic target.

**Abstract:**

O-linked β-N-acetylglucosamine (O-GlcNAc) is a reversible post-translational modification involved in the regulation of cytosolic, nuclear, and mitochondrial proteins. The interplay between O-GlcNAcylation and phosphorylation is critical to control signaling pathways and maintain cellular homeostasis. The addition of O-GlcNAc moieties to target proteins is catalyzed by O-linked N-acetylglucosamine transferase (OGT). Of the three splice variants of OGT described, one is destined for the mitochondria (mOGT). Although the effects of O-GlcNAcylation on the biology of normal and cancer cells are well documented, the role of mOGT remains poorly understood. In this manuscript, the effects of mOGT on mitochondrial protein phosphorylation, electron transport chain (ETC) complex activity, and the expression of VDAC porins were investigated. We performed studies using normal and breast cancer cells with upregulated mOGT or its catalytically inactive mutant. Proteomic approaches included the isolation of O-GlcNAc-modified proteins of the electron transport chain, followed by their analysis using mass spectrometry. We found that mitochondrial OGT regulates the activity of complexes I-V of the respiratory chain and identified a group of 19 ETC components as mOGT substrates in mammary cells. Furthermore, we observed that the upregulation of mOGT inhibited the interaction of VDAC1 with hexokinase II. Our results suggest that the deregulation of mOGT reprograms cellular energy metabolism via interaction with and O-GlcNAcylation of proteins involved in ATP production in mitochondria and its exchange between mitochondria and the cytosol.

## 1. Introduction

O-GlcNAcylation is a dynamic post-translational modification that serves as an important link between nutrient sensing, energy metabolism, and signal transduction [1]. In the cell, a small portion of glucose (about 2–4%) is utilized in the hexosamine biosynthetic pathway (HBP) to produce the nucleotide sugar uridine diphosphate-N-acetylglucosamine (UDP-GlcNAc). This metabolite is used as a substrate for the post-translational modifications of proteins, including β-O-linked N-acetylglucosaminylation (O-GlcNAcylation) [2,3]. The nutrient-sensitive O-GlcNAc cycle is regulated by two enzymes: O-linked β-N-acetyl-D-glucosaminyltransferase (OGT) catalyzes the addition of a single GlcNAc residue to a serine or threonine protein, while O-linked β-N-acetyl-D-glucosaminidase (OGA) is responsible for the removal of the GlcNAc moiety. [4]. To date, alternative splicing or alternative promotors of *OGT* result in the production of three different mRNA isoforms that code for three types of OGT enzyme. The nucleocytoplasmic OGT (ncOGT, ~116 kDa) and the short isoform of OGT (sOGT, ~78 kDa) are located in both the nucleus and the cytoplasm. The third isoform (~103 kDa) has been reported as mitochondrial OGT (mOGT) due to the presence of a mitochondrial targeting domain at its N-terminus [5]. Many studies revealed a dynamic cross-talk between phosphorylation and O-GlcNAcylation. The interplay of these two post-translational modifications may occur by steric competition for occupancy at either the same or a proximal amino acid site. The reciprocal relationship of O-GlcNacylation and phosphorylation can determine target protein functions such as subcellular localization, enzyme activity or complex formation; therefore, both modifications play an important role in the regulation of multiple biological processes [6,7].

Cancer cells reprogram their energy metabolism by increasing glucose utilization and suppressing mitochondrial respiration even in the presence of oxygen. This change, known as the Warburg effect, is a hallmark of cancer. The metabolic switch provides cancer cells with several advantages, such as precursors for the biosynthesis of nucleic acids, phospholipids, fatty acids, cholesterol, and porphyrins, which are essential for rapid growth and proliferation [8,9]. Moreover, cancer-associated abnormalities in glucose uptake promote the deregulation of cellular O-GlcNAcylation. The results of many studies suggest that increased expression of OGT and elevated O-GlcNAcylation are the common features of cancer, including breast cancer [10,11]. Breast cancer cells require O-GlcNac signaling for their initiation, progression, and metastasis [12,13]. It has been shown that the proteome of MCF-7 breast cancer cells undergoes significant changes due to altered O-GlcNAcylation by siRNA-induced downregulation of OGT. Among proteins whose expression was affected by O-GlcNAc changes, proteins involved in gene regulation, cellular localization, and cellular metabolism were identified [14]. Although there is strong evidence that O-GlcNAcylation plays a critical role in cancer cell biology, little is known about how O-GlcNAc-cycling enzymes affect mitochondrial function. Studies on the relationship between O-GlcNacylation and cancer biology have mainly focused on the function of the ncOGT isoform; in contrast, there are relatively few reports on the activity of mOGT. According to Trappannone et al. [15], ncOGT was sufficient for the attachment of O-GlcNAc moieties to mitochondrial proteins. Indeed, the vast majority of mitochondrial proteins are encoded by nuclear genes and, therefore, could be O-GlcNAcylated by ncOGT prior to translocation to mitochondria from cytosol. However, the finding that pyrimidine nucleotide carrier 1 (pnc1) functions as an UDP-GlcNAc transporter suggested that mitochondria could have the machinery to turn O-GlcNAc cycling on and off [16]. Likewise, another study found some O-GlcNAcylated proteins (cytochrome oxidase 1; MTCO1; cytochrome oxidase 2; COX2 and NADH: ubiquinone oxidoreductase core 4; MT-ND4) encoded by mitochondrial DNA [17]. Therefore, our recent report has been focused on the identification of mOGT protein substrates and the effect of mOGT dysregulation on the mitochondria activity of breast cancer cells. The results of our study showed that mOGT interacts with and modifies the proteins involved in a variety of mitochondrial processes, including transport, respiration, amino acid metabolism, protein translation, fatty acid metabolism, and apoptosis. Furthermore, we demonstrated that cellular energy metabolism and mitochondrial homeostasis are impaired by elevated mOGT expression [18]. Another study conducted on rat cardiac myocytes revealed that nearly half of the 88 proposed O-GlcNAc modified candidate proteins are associated with the oxidative phosphorylation pathway. [19]. Similarly, cardiomyocytes exposed to hyperglycemic conditions had elevated O-GlcNAcylation of respiratory chain complexes I and III and impaired mitochondria function [20]. In addition, previous reports demonstrated that some compounds of mitochondria mega-channel responses for ATP/ADP exchange are O-GlcNAcylated [21]. These observations suggest that mOGT localized on the mitochondrial inner membrane plays a pivotal role in the regulation of cellular bioenergetics. However, according to early reports, some proteins are glycosylated by both ncOGT and mOGT, while others are specifically modified by either mOGT or ncOGT [22]. The different number of tetratricopeptide repeats (TPR) in the N-terminal region of the OGT isoforms is thought to be responsible for this substrate selectivity [23]. Therefore, mOGT may have a different effect on mitochondria homeostasis than ncOGT.

Thus, in this study, we explore the effect of mOGT upregulation on respiratory chain complex activity and changes in the expression of mega-channel compounds in breast cancer cells. These results bring us closer to clarifying the effect of mOGT on mitochondrial dysfunction.

## 2. Materials and Methods

### 2.1. Reagents and Antibodies

Chemicals were purchased from Sigma-Aldrich (St. Louis, MO, USA) except as noted. Materials and reagents for cell culture were obtained from Invitrogen (Carlsbad, CA, USA), Cytogen (Sinn, Germany), and Corning Inc. (Corning, NY, USA). The used antibodies: mouse monoclonal anti-VDAC1, mouse monoclonal anti-β-actin, mouse monoclonal anti-lamin A/C, mouse monoclonal anti-phospho Ser, and mouse monoclonal anti-phospho Thr were from Santa Cruz Biotechnology, Inc., (Santa Cruz, CA, USA). The monoclonal mouse anti-O-GlcNAc (RL2) (ab2739), goat polyclonal anti-VDAC2, and rabbit polyclonal anti-VDAC3 antibodies were from Abcam (Cambridge, UK). Secondary mouse anti-rabbit (7074) as well as goat anti-mouse (7076) IgG-HRP antibodies were purchased from Cell Signaling Technology, Inc. (Beverly, MA, USA). Secondary donkey anti-goat IgG-HRP antibodies were from Abcam (Cambridge, UK). Mouse monoclonal anti-HaloTag antibodies were from Promega^®^ (Madison, WI, USA).

### 2.2. Plasmid DNA Constructs

The GeneArt service (Invitrogen^TM^) was employed to synthesize the full-length human mOGT gene reported in GenBank^TM^ (accession number U77413). The ∆CD-mOGT catalytically inactive mOGT mutant without the last 93 amino-acid region was created using the same sequence encoding mOGT (Figure 1A). The codon sequences have been normalized using GeneOptimizer^TM^ software (www.thermofisher.com) (Invitrogen^TM^) to obtain a high yield of mRNAs and proteins from synthetic genes. Following the manufacturer’s instructions, each synthesized construct flanked by Sgfl and PmeI sites was subcloned into the pFC27K HaloTag^®^ CMV-neo Flexi^®^ Vector (#G8431, Promega^TM^), then competent JM109 cells (Promega^TM^) were transformed. The fusion proteins produced by these CMV-driven constructs are tagged by the 33 kDa monomeric HaloTag protein, which is not expressed in *E. coli*, plant, or mammalian cells. The control vector that only encodes HaloTag protein was created by blunt end ligation of purified product generated after PCR amplification using designed primers and the Physion^TM^ High-Fidelity DNA Polymerase (Thermo Scientific^TM^, Rockford, IL, USA). Plasmid DNA sequences were validated following their isolation using the Extract Me Plasmid Maxi Endo-toxin-Free Kit (Blirt^®^, Gdansk, Poland).

### 2.3. Cell Culture and Treatment

Non-tumorigenic MCF10A epithelial breast cells and breast cancer cell lines MDA-MB-231 and Hs578t were purchased from the American Type Culture Collection (Manassas, VA, USA). MCF10A cells were grown in DMEM F-12 medium supplemented with 0.4% bovine pituitary extract (BPE), 3 ng/mL hEGF, 5 μg/mL insulin, and 0.5 μg/mL hydrocortisone. MDA-MB-231 and Hs578t breast cancer cell lines were grown in Dulbecco’s modified Eagle’s medium (DMEM) containing 10% (*v*/*v*) fetal bovine serum (FBS). All cell lines were cultured at 37 °C in a humidified atmosphere with 5% (*v*/*v*) CO_2_.

In experiments with different glucose availability, cells were grown for 72 h in a medium containing 5 or 25 mM glucose concentrations, which in blood correspond to normo- and hyperglycemia conditions, respectively. Cell transfection with plasmid DNA was performed using Lipofectamine^TM^ 2000 reagent (Invitrogen^TM^, ThermoFisher Scientific, Grand Island, NY, USA) according to the manufacturer’s recommendations. Cells were seeded on plates at 90% confluence and then transfected for 48 h with 1 μg of pDNA and 2 µL of Lipofectamine per well (6-well plate) according to the manufacturer’s instructions, for each experiment cells were plated in triplicates.

### 2.4. Isolation of Mitochondria-Enriched Fractions

Mitochondria were isolated from the cells using the Mitochondria Isolation Kit for Cultured Cells (Thermo Scientific) according to the manufacturer’s instructions. In the first step, 2 × 10^7^ cells were suspended in Mitochondria Isolation Reagent A and incubated on ice for 2 min. Then, the cells were crushed with 25–30 strokes in a Glass-Teflon homogenizer. The lysates were mixed with Mitochondria Isolation Reagent C and then centrifuged at 700× *g* for 10 min to remove cell debris and remaining nuclei. This step was repeated twice to remove the remaining nuclei. The supernatants were then transferred to fresh vials and centrifuged at 3000× *g* for 15 min. The mitochondria pellets were resuspended in Mitochondria Isolation Reagent C followed by centrifugation at 12,000× *g* for 10 min. The resulting mitochondria-enriched pellets were collected and frozen at −80 °C until analysis.

### 2.5. Western Blotting

Protein samples were separated using SDS-PAGE and electroblotted onto Immobilon-P transfer membranes. The quality of the transfer was verified using Ponceau S staining. The blots were then incubated with primary antibodies for 2 h at room temperature. After washing three times with Tris-buffered saline containing 0.1% Tween 20 (TBS-T), the blots were incubated for 1 h with horseradish peroxidase-coupled secondary antibodies. Immunodetection was performed with an enhanced chemiluminescence method using SuperSignal^TM^ West Pico PLUS chemiluminescence substrate (Thermo Scientific^TM^) and proteins were visualized on X-ray films or with a CCD camera. The integrated optical density (IOD) of the results was determined using Image Lab ver. 6.1 software (Bio-Rad, Hercules, CA, USA).

### 2.6. Phosphoproteins Directly Staining

Phosphoproteins were fluorescent stained on PVDF or nitrocellulose membranes using Pro-Q Diamond Phosphoprotein Blot Stain Kit (Thermo Scientific) according to the manufacturer’s instructions. The proteins electroblotted onto PVDF or nitrocellulose membrane were incubated in 7% acetic acid −10% methanol for 10 min, the membrane was washed four times in about 25 mL of dH_2_O for about 5 min. In the next step, the membrane was incubated with Pro-Q Diamond phosphoprotein blot stain reagent for 15 min. To remove excess dye from the membrane, the blot was washed three times with 30 mL of 50 mM sodium acetate −20% acetonitrile, pH 4.0. After obtaining results with the Pro-Q^®^ Diamond phosphoprotein blot stain, the blot was stained with SYPRO^®^ Ruby protein stain, to ascertain the relative phosphorylation state of proteins. The fluorescent signals were documented using the GelDoc Go Imaging system (www.bio-rad.com) (Bio-Rad).

### 2.7. Respiratory Chain Complexes Immunoprecipitation

Isolation of respiratory chain complexes from cells was performed with Immunocapture Kits purchased from Abcam^®^, (Complex I, ab109711; Complex II, ab109799; Complex III, ab109800; Complex IV, ab109801, and ATP Synthase, ab109715) according to immunocapture protocol. The purified mitochondria from cells were resuspended in PBS to 5.5 mg/mL concentration of proteins, and then supplemented with protease inhibitors. All precipitates were prepared from the same amount of mitochondrial proteins. Then, mitochondria suspensions were completely solubilized by adding n-dodecyl-β-D-maltopyranoside (lauryl maltoside) to a final concentration of 1%. The samples were mixed and incubated on ice for 30 min, then centrifuged at 72,000× *g* for 30 min at 4 °C. The collected supernatants were incubated overnight at 4 °C with 50 µL of solid beads associated with immunocapture antibodies (10 µL each of immunocapture complex I–V). After the mixing step was completed, the beads were collected by centrifugation for 1 min at 1000× *g*. Before elution, the beats were washed four times with 100 volumes of wash buffer and mixed for 5 min followed by gentle centrifugation. In the final step, all wash buffer was removed from the above beads, and the beads were resuspended in 5 volumes of urea elution buffer (4 M Urea, HCl pH 75) for 10 min. The purified complexes released into the supernatant were collected and stored at −70 °C until analyzed.

### 2.8. 2D-Electrophoresis

The respiratory chain complexes proteins’ precipitates were combined with immobilized pH gradient (IPG) rehydration buffer, which included 7 M urea, 2 M Thiourea, 2% CHAPS, 1% TBP, and 1% Bio-Lyte ampholyte (pH 3–10) from Bio-Rad, USA. The IPG strips with a 3–10 pH gradient (BioRad Laboratories, Hercules, CA, USA) were passively rehydrated in a tray with a rehydration volume of 125 µL for 18 h, and mineral oil was added to prevent evaporation. The following steps were followed for IEF: Step 1: 50 V (linear) for 4 h; step 2: 250 V (linear) for 1 h; step 3: 1000 V (linear) for 1 h; step 4: 2000 V (linear) for 1 h; step 5: 4000 V (linear) for 2 h; step 6: 4000 V (rapid) for a total of 13,000 Vh for the entire run. The IEF was conducted using the PROTEAN i12 IEF Cell from Bio-Rad. Further, the IPG strips were balanced in Equilibration Buffer 1 (BioRad Laboratories, Hercules, CA, USA) with dithiothreitol (DTT) for protein reduction and Equilibration Buffer 2 (BioRad Laboratories, Hercules, CA, USA) with iodoacetamide for protein alkylation. A second dimension separation was carried out using Criterion™ TGX Precast Gels 4–20% (BioRad Laboratories, Hercules, CA, USA). All separation parameters were followed as per the manufacturer’s instructions. Then, the resolved proteins were electroblotted onto Nitrocellulose transfer membranes (Merck Millipore Corp, Bedford, MA, USA), and the O-GlcNAc-modified proteins were detected using the GelDoc Go Imaging system (www.bio-rad.com) (Bio-Rad).

Furthermore, a bioinformatics analysis of the obtained images was performed using SameSpots software ver. 5.1.012 (TotalLab Ltd., Newcastle upon Tyne, UK). Changes in protein expression were assessed based on a ratio of 2.

### 2.9. Identification of Proteins using Mass Spectrometry Analysis

#### 2.9.1. Chemicals

The Pierce^®^ LTQ ESI positive ion calibration solution and PierceTMC18 tips were acquired from Thermo Fischer Scientific (Rockford, IL, USA). Sigma-Aldrich (St. Louis, MO, USA) provided LC-MS-grade acetonitrile, HPLC-grade formic acid, and MS Qual/Quant QC Mix. Water was purified using a Mili-Q water purification system (Millipore Corp, Bedford, MA, USA).

#### 2.9.2. Sample Preparation Prior to LC-MS/MS

Each spot was cut from the membrane and placed into 1.5 mL Eppendorf tubes. Then, 1 mL of PVP-40 solution was added to block nonspecific protein binding sites on the nitrocellulose and the spots were incubated at 37 °C for 30 min with gentle agitation using a thermomixer. Following that, the membrane was washed 8 times with 1 mL of dH_2_O to remove excess PVP-40 and transferred to a 200 µL Eppendorf tube (PCR style). On-membrane digestion was initiated by adding to each vial 20 µL of Trypsin Gold (Promega, Madison, WI, USA) solution (12.5 ng/µL in 50 mM NH_4_CO_3_, freshly diluted. As needed, a 50 mM ammonium bicarbonate buffer was added to completely cover the spots. Digestion was performed overnight at 37 °C with gentle agitation using a thermomixer. After digestion, the samples were dried using a Speed Vac centrifuge (Eppendorf, Hamburg, Germany). Then, the HPLC grade acetone was added (90 µL per 4 mm^2^ of nitrocellulose) and the vials were incubated for 30 min at RT to allow complete dissolution of the nitrocellulose and precipitation of the tryptic peptides absorbed into it. The digests were centrifuged for 10 min at 14,000× *g*, and the acetone containing the dissolved nitrocellulose was removed. Samples were dried using a Speed Vac centrifuge and re-suspended in 20 µL of 2% (*v*/*v*) acetonitrile − 0.1% (*v*/*v*) formic acid. Then, samples were purified using Pierce^TM^ C18 tips according to the provided instructions. The C18 tips were activated with 100 µL of acetonitrile and stabilized with a mixture of 50% acetonitrile and 0.1% formic acid in water. The sample (10 µL) was loaded into the Pierce^TM^ C18 tip using a pipette with 10 up-down cycles. The trapped peptides were then washed with 100 µL of a solution containing 0.1% formic acid. Finally, the sample was eluted from the Pierce^TM^ C18 tip using 20 µL of a solution containing 0.1% formic acid and 95% acetonitrile into a vial for LC-MS/MS analysis.

#### 2.9.3. LC-MS/MS Analysis

The LC-MS/MS analysis was carried out using a Transcend™ TLX-2 multiplexed LC system coupled with a Q-Exactive Orbitrap mass spectrometer (Thermo Scientific, Hudson, NH, USA) and heated electrospray ionization (HESI–II) interface. The method used was based on the protocols described by Kockmann et al. [24], Geiger et al. [25], and Velloso et al. [26], with some minor modifications. The samples were separated on a C18 Acclaim PepMapTM 100 column (1.0 × 150 mm, 3 μm particle size, nanoViper, Thermo Fisher Scientific, PA, USA) thermostated at 25°C. The mobile phases were eluent A, consisting of FA/water (0.1/99.9, *v*/*v*), and eluent B, consisting of FA/acetonitrile (0.1/99.9, *v*/*v*). The flow rate was set to 75 µL/min and the gradient used was as follows: 0–0.25 min, 2% B; 0.25–54 min, 2–35% B; 54–75 min, 35–2% B; and then 2% B was maintained for an additional 15 min for column re-equilibration. The sample injection volume was 10 μL. The Q-Exactive Orbitrap mass spectrometer was operated in positive ionization mode with full MS and all-ion fragmentation (AIF) mode. The accuracy and mass calibration were performed according to the manufacturer’s recommendations using a mixture of standards within the mass range of *m*/*z* 138.06619–1779.96528. The capillary temperature was set to 250 °C and the aux gas heater temperature to 100 °C. The electrospray capillary voltage and S-lens radio frequency (RF) level were adjusted to 2.5 kV and 50 V, respectively. Nitrogen was used as both sheath gas and auxiliary gas with flow rates of 5 and 1 (arbitrary units), respectively. The acquisition method consisted of two scan events, full MS-SIM, and AIF. The full MS-SIM scan spectra were collected within an m/z range of 350–2000, with a mass resolution of 17,000 full-width at half-maximum (FWHM) at *m*/*z* = 200. The automatic gain control (AGC) target (the number of ions to fill C-Trap) was set to 3.0 × 10^6^with a maximum injection time (IT) of 50 ms. The second scan event involved collision-induced dissociation C-trap (CID) with a normalized collision energy (NCE) of 25 V, acquired over an *m*/*z* range of 350–2000. For the AIF scan, the mass resolution was 70,000 FWHM at 100 *m*/*z* with an AGC target of 2.0 × 10^5^ and maximum IT of 100 ms. The instrument was controlled and data acquisition and evaluation were performed using Qexactive Tune 2.1, Aria 1.3.6, and Thermo Xcalibur 2.2 software, respectively.

#### 2.9.4. Analysis of Proteomic Data

Proteome Discoverer 2.4.0.305 (Thermo Fisher Scientific) was used to analyze the raw MS/MS data. The SEQUEST HT algorithm was employed to search the MS/MS spectra against a human FASTA-formatted database (SwissProt, v2017-10-25 with taxonomy Homo sapiens, 42,252 sequences). The precursor mass tolerance was ±20 ppm and the fragment ion mass tolerance was ±0.6 Da. Trypsin was chosen as the enzyme specificity. The minimum peptide length was set to six amino acids and a maximum of two missed cleavages were allowed. Database searches included carbamidomethylation on cysteine as a static modification, and oxidation on methionine and acetylation of protein N-terminal as dynamic modifications. To identify peptides and proteins, a false discovery rate (FDR) with a target-decoy strategy was set to 0.01.

### 2.10. Cell Imaging

Cells were plated for 24 h at a density of 1.5 × 10^4^ cells/well and then treated with pmOGT-HaloTaq or pHaloTaq as described in the cell culture and treatment section. 48 h after transfection, the medium of the cells was replaced for 30 min with medium containing cell-permeable HaloTaq^®^ Coumarin Ligand (Promega^TM^). The medium was then replaced for 30 min with an equal volume of fresh, warm culture medium containing the cell-permeant dye Mito Tracker^TM^ Red FM (Thermo Scientific^TM^), which accumulates in active mitochondria in living cells. To remove excess dye from the cells, the monolayers were washed twice with fresh warm culture medium for 15 min. DiOC18(3), which intercalates into the plasma membrane, was used to stain living cells. The fluorescence emission signals were recorded at 25 °C and 63× magnification using a Leica TCS SP8 system.

### 2.11. Respiratory Chain Complex Activity Measurement

The cells were seeded in a 15 cm dish with approx. 1.4 × 10^7^ cells for the MCF-10A and Hs578t cell lines and 1 × 10^6^ cells for the MDA-MB-231 cell line to reach a confluence of approx. 80 after 24 h. The cells were then treated with pmOGT HaloTaq or pHaloTaq for the next 48 h. At harvest time, cells were washed twice with ice-cold PBS, scraped in PBS, pelleted at 1000× *g*, and used to isolate mitochondria using the Mitochondria Isolation Kit for Cultured Cells (Thermo Scientific^TM^). The purified mitochondria were suspended in 400 µL of buffer C and lysed for 30 min with detergents by adding lauryl maltoside to a final concentration of 1%. After centrifugation, the mitochondrial protein was quantified using the Lowry method and the mitochondrial lysates were adjusted to the same concentration of mg/mL before loading into the microplate assay.

#### 2.11.1. Complex I Enzyme Activity

The Abcam Complex I Enzyme Activity Kit (ab109721) was used to assess mitochondrial complex I activity, following the manufacturer’s instructions. The activity was determined spectrophotometrically by measuring the oxidation of reduced NADH to NAD+ and the simultaneous reduction of a dye, resulting in an increase in absorbance at OD = 450 nm. Equal amounts of each sample in triplicate were added to the plate. Colorimetric intensity was quantified using a Bio-Tek Synergy HT Microplate Reader (Bio-Tek Instruments, Winooski, VT, USA) at 450 nm, with kinetic readings taken every 30 s for 1 h. The activity of the complex is expressed as change in absorbance per minute (mOD/min).

#### 2.11.2. Complex II Enzyme Activity

The activity of oxidative phosphorylation complex II was determined using the Abcam Complex II Enzyme Activity Kit (ab109908) according to the manufacturer’s instructions. The assay evaluates activity by the decrease in absorbance at OD600 nm caused by the reduction of DCPIP (2,6-diclorophenolindophenol) and the production of ubiquinone. 50 µL of a sample with a protein concentration of 5 µg/µL was added to each well of the microplate in triplicate. Absorbance at 600 nm was measured for 1 h using a 20-s interval kinetics program using a Bio-Tek Synergy HT Microplate Reader (Bio-Tek Instruments, Winooski, VT, USA). The activity of the complex is expressed as change in absorbance per minute (mOD/min).

#### 2.11.3. Complex III Enzyme Activity

Mitochondrial complex III activity was assessed using the Abcam Mitochondrial Complex III Activity Assay kit according to the manufacturer’s protocol. The activity was determined as a absorbance at 550 nm of reduced cytochrome c. Colorimetric intensity was quantified using a Bio-Tek Synergy HT Microplate Reader (Bio-Tek Instruments, Winooski, VT, USA). The activity of the complex is expressed as change in absorbance per minute (mOD/min).

#### 2.11.4. Complex IV Enzyme Activity

Abcam’s Complex IV Enzyme Activity Microplate Assay Kit (ab109909) was used for the analysis of mitochondrial OXPHOS Complex IV enzymatic activity. Activity was determined colorimetrical by following the oxidation of reduced cytochrome c as an absorbance decrease at 550 nm. The assays were read using a Bio-Tek Synergy HT Microplate Reader (Bio-Tek Instruments, Winooski, VT, USA). The activity of the complex is expressed as change in absorbance per minute (mOD/min).

#### 2.11.5. ATP Synthase Complex Activity

Mitochondrial Complex V activity was measured using ATP Synthase Specific Activity Microplate Assa Kit (ab109716, Abcam) according to the manufacturer’s instructions. In the first step of this assay, immunocaptured ATP synthase within the wells hydrolyzed ATP to ADP and phosphate. The production of ADP was linked to the oxidation of NADH to NAD+. This oxidation was monitored by measuring the decrease in absorbance at 340 nm. The quantity of enzyme was then measured in the same wells by adding a Complex V specific antibody conjugated with alkaline phosphatase. The enzyme catalyzes a reaction that results in a color change of the substrate from colorless to yellow. The increase in absorbance at 405 nm in time was proportional to the amount of protein captured in the well. Colorimetric intensity was quantified using a Bio-Tek Synergy HT Microplate Reader (Bio-Tek Instruments, Winooski, VT, USA). The activity of the complex is expressed as change in absorbance per minute (mOD/min).

### 2.12. RT-PCR

Total RNA was isolated from breast cancer cells using the Total RNA Isolation Kit (A&A Biotechnology, Gdynia, Poland) following the manufacturer’s instructions. To obtain first-strand cDNAs, 1 μg of total RNA was reverse transcribed using the High-Capacity cDNA Reverse Transcription kit (Applied Biosystems, Foster City, CA, USA). The expression of the VDACs genes was analyzed using the TaqMan Gene Expression Assay (Applied Biosystems). The inventoried assays used for VDAC1, VDAC2, VDAC3, and the internal control glyceraldehyde-3-phosphate dehydrogenase (GAPDH) included fluorogenic, FAM-labeled probes and sequence-specific primers (Hs04978484, Hs01075603, Hs00366592, and Hs99999905, respectively). The reactions were performed in triplicate using Eppendorf RealPlex thermalcycler. The abundance of studied mRNA genes in samples was quantified using the ΔC method. Ct (Ct_gene_–Ct_GAPDH_); values were recalculated into relative copy number values (number of copies of gene of interest mRNA per 1000 copies of reference gene mRNA).

### 2.13. ATP Level Measurement

The ATP level was quantified using a Luminescent ATP Detection Assay Kit (ID: ab113849; Abcam^®^, Cambridge, UK) in accordance with the manufacturer’s instructions. Initially, cells were plated in a 96-well plate (Clear^®^, white, flat bottom; Greiner Bio-One GmbH, Frickenhausen, Germany) and grown for one day before treatment. After transfection with plasmid DNA for 48 h, the ATP level was measured. Cell lysates were mixed with ATP detection reagent for 10 min at room temperature. The luminescent signal was measured with a plate reader (SpectraMax^®^ PARADIGM^®^ Multi-mode Microplate Plate Reader; Molecular Devices, San Jose, CA, USA). The results were presented as a percentage of ATP levels relative to the control cells.

### 2.14. Statistical Analysis

Statistical evaluation was performed with Statistica ver. 13 (StatSoft Inc., Tulsa, OK, USA) data analysis software. Comparison between groups was performed using Student’s paired *t*-test. A *p* value of <0.05 was considered significant.

## 3. Results

### 3.1. mOGT Upregulation in Breast Cells Decreases Mitochondrial Phosphorylation as a Result of Increased O-GlcNAcylation

The primary function of OGT is to catalyze the attachment of the O-GlcNAc residue to proteins. However, recent evidence suggests that OGT may also have non-catalytic functions, and some discrepancies have been observed between the genetic knockout of OGT and the inhibition of its catalytic activity [27]. A recent study by Levine Z et al. [28] showed that catalytically inactive OGT can cooperate within the same cellular pathway to alter cell physiology and is sufficient to drive mouse cell fibroblast proliferation. They found that the non-catalytic functions of OGT are related to oxidative phosphorylation and components of the actin cytoskeleton. To exclude the non-catalytic effect of mOGT on respiratory chain activity and to identify mOGT respiratory chain candidate protein substrates, we developed plasmids containing the complete mOGT sequence and a mutant version lacking the second catalytic domain (CDII). These sequence constructs were inserted into an empty vector that allows them to be expressed as fusion proteins with HaloTaq at the C-terminus in both normal and cancerous breast cells (mOGT-HaloTaq and ΔCD-mOGT-HaloTaq; for the full-length and its mutant with truncated catalytic domain, respectively) (Figure 1A). As previously reported, enhanced expression of catalytic active mOGT promotes programmed cell death probably due to the mOGT cytotoxic effect [29]. This is the main reason for the failure to achieve a stable cell line overexpressing active mOGT. Therefore, our study was conducted using the transient expression of mOGT-HaloTaq or ΔCD-mOGT-HaloTaq proteins versus only HaloTag protein as a control. Since strong over-expression of mOGT in cells can lead to cell apoptosis and endogenous mOGT levels are dependent upon glucose concentration, we tested different doses of plasmid DNA for treatment cells growing in different glucose conditions. In all experiments, we used the same dose of pDNA, sufficient to increase mOGT expression in the mitochondria of cells while not affecting cell viability (Figure 1B,C and Appendix A). As shown in Figure 1D, cells transfected for 48 h with 1 µg pDNA/ per ml of culture medium followed by treatment with G418 had similar expression of mOGT compared to cells growing without a selective pressure marker. As expected, in cells transfected with pmOGT-HaloTaq, mOGT-HaloTaq fused protein was colocalized with Mito Tracker Red, a selective mitochondria membrane marker (Figure 1E and Appendix A).

According to the published data, O-GlcNAc and phosphate can modify the same amino acids reciprocally or amino acids proximal to each other [6,7], so that an increase in mOGT could inhibit the phosphorylation of some mitochondrial proteins to regulate their function. To determine whether catalytically active mOGT affects the mitochondrial phosphorylation state of mammary cells, we upregulated OGT expression in cells grown under normal and high-glucose conditions by transfecting them with plasmids encoding mOGT versus the vector encoding only HaloTag as a control. First, we examined the effects of increased mOGT expression on mitochondrial O-GlcNAcylation status. Western blotting was used to detect the relative O-GlcNAc content of mitochondrial proteins, followed by densitometric analysis using the Rl2 antibody. We observed that the upregulation of full-length mOGT led to an increase in O-GlcNAc-modified proteins in all cell lines. The most pronounced changes in O-GlcNAc levels were seen in cells cultured under low-glucose conditions (Figure 2A). It is important to note that the level of O-GlcNAcylation of mitochondrial proteins depends not only on mOGT but also on ncOGT and OGA. Our previous research [18] has shown that changes in mOGT expression do not affect the level of OGA and ncOGT expression in breast cancer mitochondria. Additionally, silencing ncOGT using siRNA did not significantly impact O-GlcNAcylation levels in the mitochondrial fraction. Therefore, it appears that ncOGT does not play a major role in the O-GlcNAcylation of mitochondrial proteins in breast cancer cells. To determine the influence of mOGT on the phosphorylation state of mitochondrial proteins, we stained electrophoretically resolved proteins with the fluorescent reagent Pro-Q Diamond, which selectively binds to the phosphate moiety. Interestingly, the effect of increased mOGT levels on mitochondrial protein phosphorylation, as opposed to O-GlcNAcylation, was slightly better in cells grown under high-glucose conditions than under low-glucose conditions. The decrease in mitochondrial protein phosphorylation in pmOGT-HaloTaq-treated cells compared to control cells was 13% in MCF-10A cells, 40% in MDA-MB-231 cells, and 23% in Hs 578t cells (Figure 2B). The most frequently phosphorylated amino acids are serine, threonine, tyrosine, and histidine, but O-GlcNAcylation competes with phosphorylation only for the modification of serine and threonine residues [6,7]. Therefore, immunodetection was used to determine the relative p-Ser/p-Thr levels of mitochondrial proteins in cells treated under the same conditions. The results of densitometric analysis showed that mOGT inhibited up to 34% of mitochondrial phosphoproteins in the cell line MDA-MB-231 grown under low-glucose conditions and up to 42% of mitochondrial phosphoproteins in Hs578t cells grown under high-glucose conditions (Figure 2C).

Taken together, these observations suggest that the upregulation of mOGT effectively increases the O-GlcNAcylation level of mitochondrial proteins and limits mitochondrial protein phosphorylation in breast cells.

### 3.2. Increased mOGT Expression Affects Respiratory Chain Complex Activity

There is growing evidence that O-GlcNacylation regulates energy metabolism through the dynamic modification of mitochondrial proteins and that changes in O-GlcNAc-forming enzymes are associated with mitochondrial dysfunction. A recent study revealed that elevated mitochondrial O-GlcNAcylation is associated with the impaired activity of complex I, III, and IV in cardiomyocytes under hyperglycemic conditions [20]. Given these findings, our study aimed to find out whether the expression of mOGT and its catalytic activity affects the activity of mitochondrial respiratory chain complexes in breast cells. To exclude the influence of glucose concentration on the conclusion, OGT expression was upregulated by the transfection of plasmids encoding either mOGT or the catalytic dead version compared to the vector encoding only HaloTaq as a control in cells grown under normal and high-glucose conditions. The activity of the individual oxidative phosphorylation complexes was assessed by colorimetric and fluorometric microplate assays. Cells transfected with pmOGT-HaloTaq showed a significant increase in Complex I compared to the control cells (cells transfected with pHaloTag) (Figure 3A). Similarly, increasing the expression of full-length mOGT or catalytically inactive mOGT in all cell lines resulted in a significant increase in Complex II activity (Figure 3B). The activity of Complex III was slightly increased only in MDA-MB-231 cells grown under low-glucose conditions, whereas there were no changes in MCF-10A cells (Figure 3C). In contrast, cells that upregulated mOGT showed markedly reduced activity of complex IV compared to control cells, with the most altered activity observed in cancer cell lines (Figure 3D). Finally, we determined the effect of elevated mOGT expression on the activity of ATPase, which synthesizes ATP from ADP and inorganic phosphate using exergonic proton reflux. We observed that in both normal and cancerous breast cells the increase in the full-length or catalytically inactive mOGT expression led to a significant decrease in Complex V activity (Figure 3E). Hence, these findings imply that mOGT controls the rate of oxidative phosphorylation and its up-regulation limits mitochondrial ATP production regardless of glucose availability.

### 3.3. mOGT Modify Some Electron Transport Chain Proteins

Our recent study proved that mOGT has catalytic activity and is able to O-GlcNAcylate a myriad of mitochondrial proteins in breast cancer cells. Based on the proteomic approach we indicated a group of 107 candidate proteins as mOGT substrates and interacting partners [18]. In another study, Hu and collaborators [20] identified 88 O-GlcNacylated mitochondrial candidate proteins in rat cardiac myocytes, almost half of which are components of the oxidative phosphorylation system. In light of this evidence and our observations, there is a possibility that a variety of components of the electron transport chain (ETC) are altered by mOGT. To identify the proteins of the respiratory chain complexes O-GlcNAcylated by mOGT, we enriched mitochondria from normal and breast cancer cells transfected with pmOGT-HaloTaq or pHaloTaq vectors and then used the immunocapture kits to purify the proteins of the ETM complexes. The effect of mOGT upregulation and pull-down efficiency were verified using Western blotting. The results showed that the increased mOGT expression elevated the O-GlcNAcylation of the ETC proteins, while their phosphorylation level was decreased, especially in cancerous breast cells (Figure 4A,B, Appendix A). The isolated precipitates were resolved by 2D electrophoresis, and the O-GlcNAc-modified proteins were subsequently immunodetected with RL2 antibodies. Cut spots were selected by comparative analysis of O-GlcNAcylated proteins between samples that were transfected with pmOGT-HaloTaq and pHaloTaq. Candidate proteins were identified by nano-LC-MS/MS as described in the methods. Changes in protein expression were assessed based on a ratio of 2. We have identified a group of 162 candidates that are specifically modified by mOGT (Figure 4C). In the second step, we compared the mOGT candidate substrates identified in our study with the mitochondrial ETC components listed in the MitoCart 3.0 datasets (Broad Institute, Cambridge, MA, USA), as well as with the O-GlcNAcylated proteins listed in the Human O-GlcNAcome Database ver. 1.3 (Medical College of Wisconsin, Milwaukee, WI, USA). Based on these, we found out the group of 125 candidates as ETC components, and 62 of them were previously described as O-GlcNAcylated. Some of the candidates were identified in only one cell line, but 70 proteins were found in both normal and cancerous breast cells and 82 candidates were found in at least two of the three cell lines analyzed (Figure 4 D). Among them, a set of 43 proteins was previously reported to be O-GlcNAc-modified (Figure 4E). Finally, to narrow the gold list of mOGT-modified ETC substrates, we prepared a list of 19 proteins that fulfilled the following criteria: (1) were present in all three cell lines, (2) were characterized as ETC components according to MitoCart 3.0 datasets, and (3) were previously reported as O-GlcNAc modified according to the Human O-GlcNAcome Database (Table 1). All identified mOGT ETC substrate candidates are presented in Appendix A.

### 3.4. Elevated mOGT Impact on VDAC Porins and ATP Level

According to published data, impaired mitochondrial function is associated with an increase in the O-GlcNAcylation of mitochondrial proteins. More specifically, O-GlcNAcylation regulates oxidative phosphorylation in several steps and controls ATP generation [17,19,20]. However, cytoplasmic ATP levels are controlled by both ATP production rates and ATP exchange between mitochondria and cytoplasm. ATP is mainly transported into the cytoplasm by VDAC (ang. voltage-dependent anion channel) porins, in particular by the most abundant VDAC1 isoform [30]. To determine whether mOGT expression affects the level of mitochondrial VDAC channels of normal breast and cancer cells, OGT expression was upregulated in cells grown under normal-glucose conditions by transfection with plasmids encoding mOGT compared to the vector encoding only HaloTaq as a control. First, we used a real-time PCR method to determine the effects of increased mOGT expression on the level of VDACs mRNA. MCF-10A cells transfected with pmOGT-HaloTaq showed a significant decrease in VDAC1, VDAC2, and VDAC3 mRNA levels, whereas in MDA-MB-231 and Hs578t breast cancer cells only VDAC3 and VDAC1 mRNA levels decreased significantly after mOGT upregulation (Figure 5A). To confirm the effect of mOGT on VDACs expression, we determined VDACs protein levels in the mitochondria of cells treated with pmOGTHaloTaq and the control plasmid. Despite the decrease in VDAC1 and VDAC2 mRNA, we observed that mOGT did not affect the levels of VDAC1 and VDAC2 proteins in the mitochondria of normal and breast cancer cells. In contrast, an increase in mOGT in all cell lines led to a significant decrease in VDAC3 porin in the mitochondria (Figure 5B, Appendix A). VDAC3 exhibits cysteines predicted towards the intermembrane space; therefore, this isoform is preferred target for oxidation by ROS. These properties suggest that VDAC3 could serve as a marker for redox signaling in the mitochondrial intermembrane space [31,32].

To date, it is known that VDACs porins regulate signaling pathways that contribute to glycolysis rate, cell survival, and apoptosis. In particular, complex formation between the most abundant VDAC1 and hexokinase-II (HK II) is crucial for cell growth at high glycolysis rate. Interestingly, VDAC1 is also a substrate of glycogen synthase kinase (GSK3β) and its phosphorylation by GSK3β is associated with reduced HKII binding [33]. Our recent proteomic study confirmed that the VDAC1 porin is O-GlcNAcylated by mOGT, so the upregulation of mOGT could alter the interaction of VDAC1 with kinases. Therefore, we used co-immunoprecipitation to determine the influence of mOGT on the binding of HKII and GSK3β to VDAC1. In all cell lines, the upregulation of mOGT led to an attenuation of HKII binding to VDAC1 and this effect was independent of HKII levels in the mitochondria. In contrast, mOGT-related O-GlcNAcylation had no effect on the association of GSK3β with VDAC1 (Figure 5B and Appendix A).

Finally, we measured the intracellular levels of ATP in cells transfected with pmOGT-HaloTaq and pHaloTaq, grown under normal-glucose concentration. In both normal and breast cancer cells, elevated mOGT resulted in a decrease in intracellular ATP levels compared to control cells (see Figure 5C).

Together, these observations suggest that the upregulation of mOGT may limit the intracellular ATP production in two different ways: by inhibiting ATP generation in oxidative phosphorylation and as a result of reducing the rate of glycolysis by impairing HKII binding to VDAC1.

## 4. Discussion

Increased O-GlcNAcylation of cellular proteins is a characteristic feature of many types of cancer, including breast cancer. O-GlcNAcylation, considered as cellular metabolic sensor is closely linked to HBP flux, which is supplied by diverse nutrient sources including sugars, fatty acids, and amino acids. Therefore, one of the main reasons for impaired O-GlcNAcylation of proteins is altered glucose metabolism, particularly an accelerated rate of glycolysis. Notably, cancer cells are able to reprogram their glucose metabolism and favor ATP production in glycolysis even under aerobic conditions. Thus, accelerated glucose uptake and utilization in cancer cells result in enhanced glucose flux through HBP and increased O-GlcNAcylation [1,2,3,4,5]. The second major reason for elevated O-GlcNacylation in cells is the impaired level of enzymes that catalyze the attachment and removal of O-GlcNAc residues. Increased OGT expression and decreased in OGA expression have been reported in a number of human cancers [10,12]. Recent evidence indicates that OGT and OGA are reciprocally controlled at the transcriptional level to maintain a balance of O-GlcNAc in healthy cells, but in cancer cells, the upregulation of both enzymes results in the promotion of tumor growth [34]. Therefore, the O-GlcNAcylation pattern may be varied in normal and cancer cells. Indeed, Champattanachai et al. [11] compared the O-GlcNAcylation pattern in normal tissue and breast cancer tissue using 2D O-GlcNAc immunoblotting and LC-MS/MS analysis. They identified a group of 29 proteins as potential substrates for O-GlcNAc, out of which 7 were found to be specifically O-GlcNAcylated in breast cancer. Some of these proteins were associated with the Warburg effect. These included metabolic enzymes, stress response proteins, and biosynthesis-related proteins. Another study identified 155 O-GlcNAcylated proteins of which 41 were only observed in lymph node metastasis breast carcinomas [13]. The modification of proteins by O-GlcNAcylation controls target protein functions at diverse mechanisms including enzymatic activity, stabilization, subcellular trafficking, or complex formation, and subsequently modulating metabolic and signaling networks [35]. Until now, understanding the relationship between impaired O-GlcNacylation and the development and progression of breast cancer was focused on the processes occurring in the cell nucleus and cytoplasm, with limited reports on the role of this modification in the mitochondria. Moreover, the role of mOGT in the O-GlcNAcylation of mitochondrial proteins and its effect on mitochondrial processes remains insufficiently understood. The previous study performed by Trapannone and colleagues [15] suggest that mOGT is present in cells temporarily and only under certain conditions, while ncOGT is the main expressed isoform capable of O-GlcNAcylation on mitochondrial proteins. According to the literature, it is suggested that the majority of mitochondrial proteins are synthesized in the cytoplasm and may undergo modification by ncOGT before being targeted to the mitochondria [15]. However, mOGT may interact with other proteins than ncOGT isoform [22]. Additionally, our recent study showed that breast cancer cells express catalytically active mOGT, and its expression is regulated by glucose, probably on the transcription level. By using proteomics-based approaches we identified a group of over 600 proteins as mOGT-interacting partners and mOGT-mediated O-GlcNAc-modified substrates. These previously reported results indicate that the increased mOGT expression caused a decrease in glycolytic rate and oxygen consumption in breast cancer cells [18]. Therefore, to explore, how mOGT contributes to energy reprogramming in breast cells, we continued our study using designed plasmids encoding the catalytically active fused mOGT and the catalytically inactive mutant displaying a deletion of the second domain. According to the initial report on mOGT function, its overexpression in INS-1 cells leads to cell cytotoxicity and apoptosis [29]. Therefore, as a starting point of this work, we tested whether the transient upregulation of mOGT can target mitochondria with low cytotoxicity of breast cells. For each cell line, the used plasmid transfection conditions had no significant impact on cell viability as well as cells co-treated with G418 had similar expression of mOGT compared to cells growing without a selective pressure marker. According to the published data, O-GlcNAcylation competes with phosphorylation for the modification of serine and threonine residues [6,7]. Our study is the first to report that mOGT-mediated increase in the O-GlcNAcylation of mitochondrial proteins led to a decrease in mitochondrial protein phosphorylation. The summarized proteomic studies revealed that many of the identified phosphoproteins in the mitochondria of mammalian tissues are involved in oxidative phosphorylation, TCA cycle, and lipid metabolism [36]. In accordance with that, our results showed that increased mOGT expression inhibits the phosphorylation of respiratory chain complexes protein in breast cells. Therefore, we applied proteomic approaches to identify electron transport chain proteins specifically modified by mOGT. The results allowed us to specify a group of 125 candidates as ETC components according to MitoCart 3.0. database and 62 of them were previously described as O-GlcNacylated according to the Human O-GlcNAcome Database ver 1.3. The identified O-GlcNAcylated ETC proteins in normal breast cells (MCF-10A) in 51% and 72% overlap with the list of O-GlcNAcylated ETC proteins in MDA-MB-231 and Hs578t breast cancer cells, respectively. Some of the candidates were identified in only one cell line, but 82 proteins were found in at least two of the three cell lines, and among them, a set of 43 proteins was previously characterized as O-GlcNAcylated. It should be noted that we did not expect a greater coverage of candidates identified between cell lines due to the differences in proteomes. However, the overlapping of identified protein lists between the different cell lines reinforces the specificity of our strategy. Finally, we identified a group of 19 candidates that were present in each of the three cell lines and were both identified as ETC components and O-GlcNAc-bearing proteins. Our analysis revealed that mOGT modifies the components of all respiratory chain complexes and is in agreement with recently published study in which isolated cardiac mitochondria were acutely exposed to the OGT substrate UDP-GlcNAc in the presence or absence of the OGA inhibitor NButGt. The study revealed that, among the multi-proteins complexes of the oxidative phosphorylation machinery, complex I (23 subunits, 60%,), V (8 subunits, 50%), III (6 subunits, 75%), and II (3 subunits, 75%), were predominantly affected with 50–75% of their subunits being significantly more O-GlcNAcylated in response to NButGT compared to only 25% (4 subunits) for complex IV [37]. The mOGT-mediated changes in the O-GlcNAcylation and phosphorylation of ETC components are associated with altered respiratory chain complex activity in breast cells. Our results showed that upregulated mOGT in breast cells led to significant increases in ETC Complex I and II activity, whereas the activity of Complex IV and V was markedly decreased. Previously, the correlation between elevated mitochondrial O-GlcNAcylation and impaired activity of complexes I, III, and IV was observed in cardiomyocytes growing under hyperglycemic conditions [20]. In another study, in cardiac mitochondria exposed to NButGT, the activity of complex I was increased by ~50%, while that of complex II and complex IV was unchanged [37]. In the recent study performed by Akinbiyi and collaborators [38] on the transformed OGA knockout (KO) mouse embryonic fibroblasts (MEFs) that exhibit a global hyper-*O*-GlcNAcylated cellular environment, complexes with diminished protein content (I, II, and IV) all exhibited decreased average enzyme activity. Our results are in agreement with previously observed increases in mitochondrial transmembrane potential and ROS generation followed by mOGT upregulation in breast cancer cells [18]. According to the literature, respiratory complexes I, II, and III are considered the main sources of mitochondrial ROS [39]. Therefore, increased activity of ETM complexes I and II may be associated with enhanced ROS generation. Interestingly, mOGT upregulation decreases VDAC3 expression in the mitochondria of breast cells. VDAC3 porin protects mitochondria from oxidative stress and excessive oxidation by ROS leads to its ubiquitination and degradation [40]. Thus, it seems that mOGT may cause a decrease in VDAC3 expression by altering the activity of respiratory complexes. Therefore, we suppose that a decrease in VDAC3 levels may intensify the cytotoxic effects of mOGT and facilitate the elimination of defective mitochondria. On the other hand, in breast cells with elevated mOGT, we observed a decrease in ATP levels probably due to decrease in the activity of complexes IV and V. In addition, our study revealed that mOGT regulates intracellular ATP levels in at least two different mechanisms associated with oxidative phosphorylation and the interaction of VDAC with hexokinase II. The O-GlcNAcylation of VDAC has been previously reported [21]; however, the effect of this modification on mitochondrial porin function was not elucidated. Our study is the first to report on the altered interaction of VDAC1 with HKII as an effect of mOGT upregulation. HKII binding to VDAC1 has direct access to ATP and the glycolytic enzyme is then less sensitive to inhibition by glucose-6-phosphate [41]. Therefore, in cells with elevated mOGT decreased intracellular ATP levels may be caused by the decreased activity of hexokinase II. Although mOGT seems to be implicated in the regulation of glycolytic rate in breast cells, this hypothesis should be validated in the future with an additional approach based on the RNA interference method.

## 5. Conclusions

Protein O-GlcNAcylation is crucial for regulating various biological processes. In this study, we focused on the role of poorly explored mitochondrial OGT isoform in the regulation of cellular bioenergetics. We found that the upregulation of mOGT in normal and breast cancer cells impacts protein phosphorylation in mitochondria. Increased mOGT expression affects the activity of mitochondrial respiratory chain complexes and alters the interaction of VDAC porins with kinases. It seems that mOGT deregulation may be responsible for changes in cellular energy metabolism via interaction with and the O-GlcNAcylation of proteins involved in ATP production in mitochondria and its exchange between mitochondria and cytosol.

## Figures and Tables

**Figure 1 cancers-16-01048-f001:**
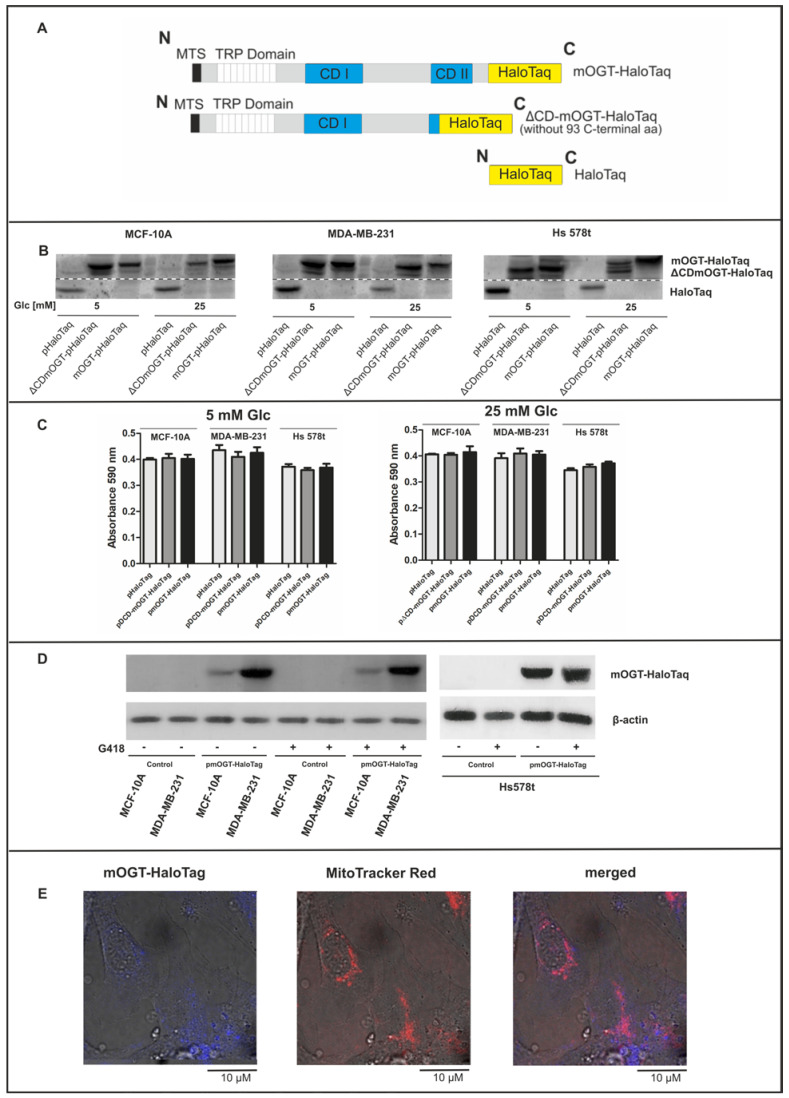
Transient mOGT upregulation is sufficient to targets the mitochondria without cytotoxic effects. (**A**) The scheme of mOGT-HaloTaq fused protein, its catalytic inactive mutant with truncated CDII domain, and HaloTaq used as a control. (**B**) Expression of Halo-tagged proteins in mitochondria-enriched fraction 48 h after transfection of cells growing in low or high-glucose conditions. (**C**) Viability of cells determined by MTT assay. Cells were growing in low and high-glucose conditions and treated for 48 h with plasmid DNA. The bars represent mean +/− SD from three independent experiments in triplicate. (**D**) Expression of HaloTaq fused proteins 72 h after plasmid transfection in cells growing in medium with or without G418 selective pressure marker; (**E**) The confocal images of MCF-10A cells stained with cell-permeable HaloTaq^®^ Coumarin Ligand (bleu) counterstained with Mito TrackerTM Red FM (red) showed effective targeting mitochondria by mOGT (images taken at 63× magnification).

**Figure 2 cancers-16-01048-f002:**
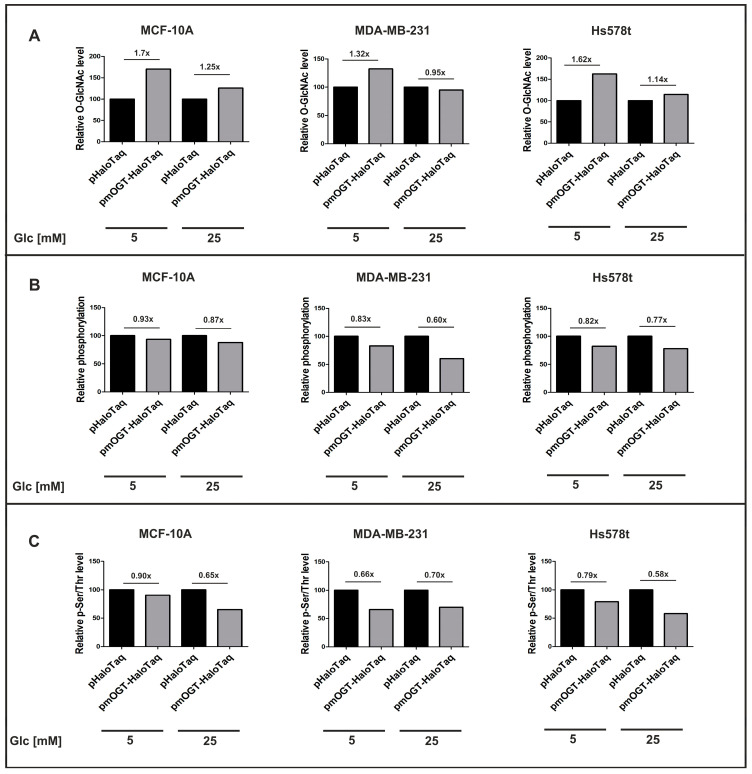
mOGT-mediated increase in O-GlcNAcylation led to a decrease in the phosphorylation of mitochondrial proteins. (**A**) Densitometric analysis of O-GlcNAc levels in mitochondria-enriched fraction from Western blots; (**B**) densitometric analysis of phosphorylated protein levels in mitochondria determined by Pro-Q Diamond phosphoprotein blot stain reagent; (**C**) densitometric analysis of p-Ser/pThr levels in mitochondria from Western blot. The analyses were performed using Image Lab ver. 6.1 software (Bio-Rad).

**Figure 3 cancers-16-01048-f003:**
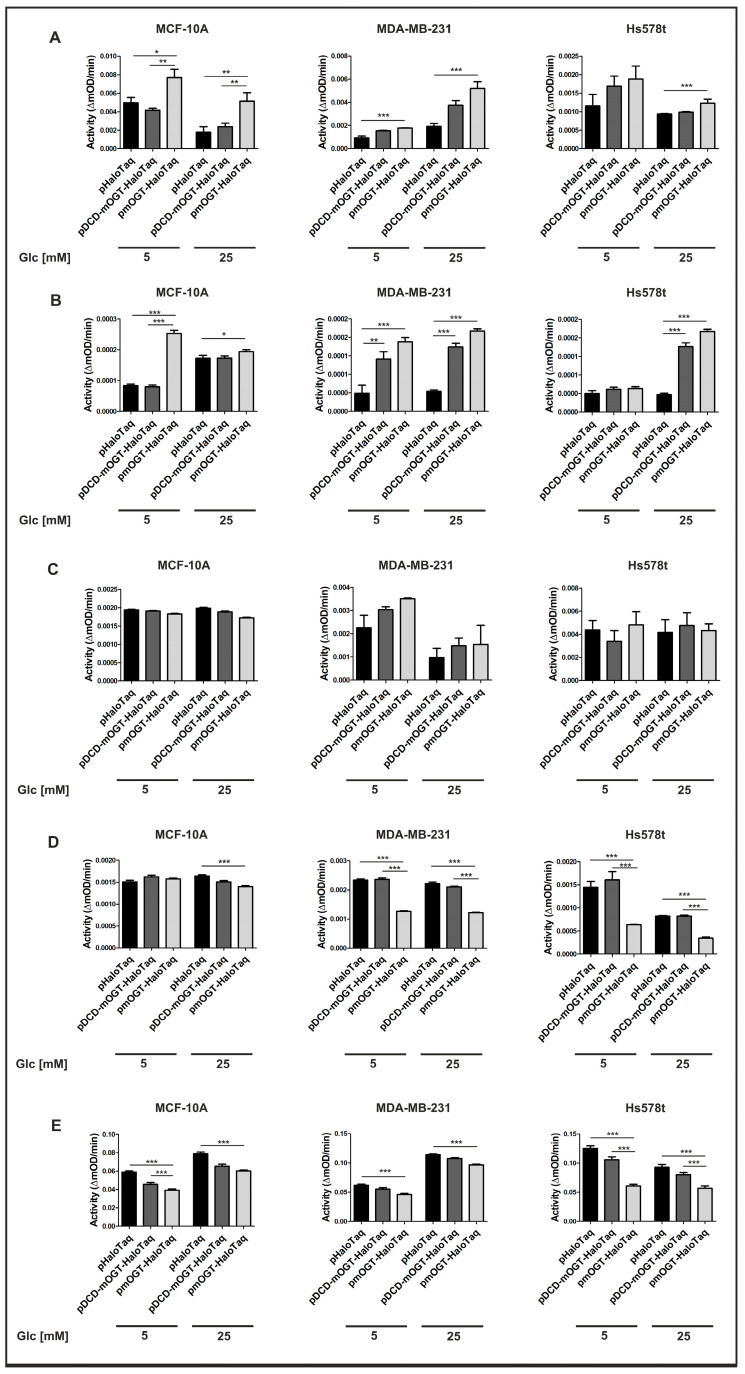
Upregulated mOGT alters the activity of respiratory chain complexes. (**A**) Complex I; (**B**) Complex II; (**C**) Complex III; (**D**) Complex IV; (**E**) ATPase synthase complex. The activity of the complexes is expressed as a change in absorbance per minute (mOD/min). Data represent the average of at least 3 independent experiments performed in duplicates and are presented as a means +/− SD. * indicates significance * *p* < 0.05; ** *p* < 0.01; *** *p* < 0.001.

**Figure 4 cancers-16-01048-f004:**
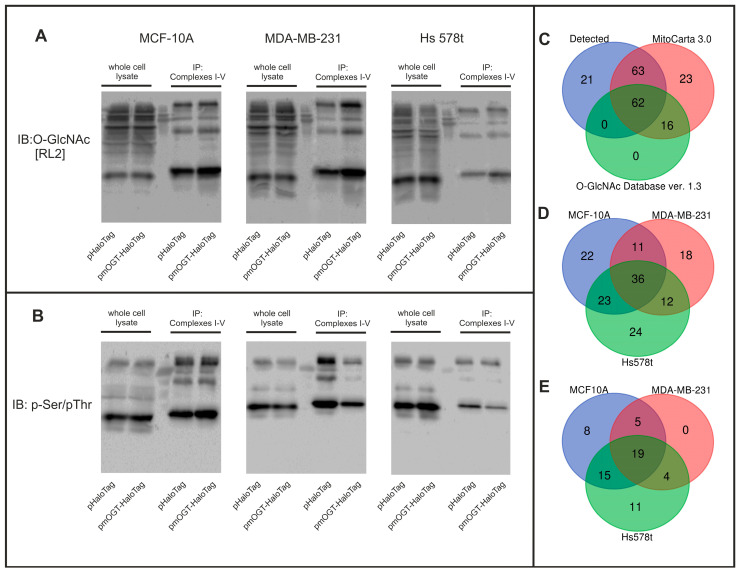
Elevated levels of mOGT reduce phosphorylation of respiratory chain proteins and mediate their O-GlcNAcylation. (**A**) O-GlcNAcylation levels of ETM components determined by Western blot; (**B**) p-Ser/p-Thr expression of isolated ETM proteins determined by Western blot; (**C**) Venn diagram showing the total number of identified ETM mOGT candidates common with ETM components according to the MitoCart 3.0 datasets (Broad Institute) and O-GlcNAcylated proteins according to the Human O-GlcNAcome Database ver. 1.3 (Medical College of Wisconsin); (**D**) The graph presents identified ETM mOGT candidates in three analyzed cell lines; (**E**) Venn diagram showing mOGT candidates common to all analyzed cell lines previously identified as ETM O-GlcNAcylated proteins according to the MitoCart 3.0 datasets and the Human O-GlcNAcome Database ver. 1.3.

**Figure 5 cancers-16-01048-f005:**
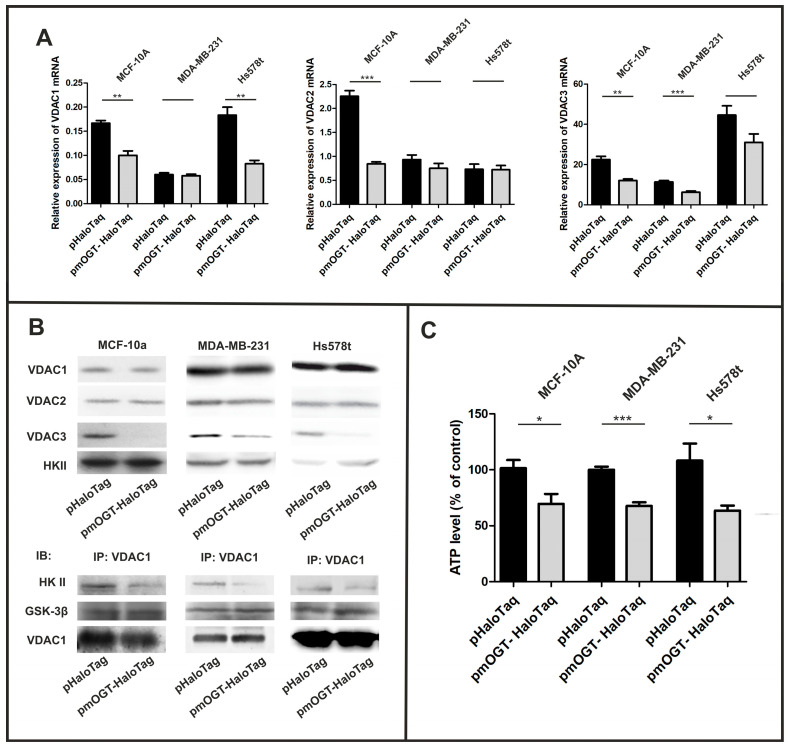
Impact of mOGT upregulation on expression and function of mitochondrial VDAC porins. (**A**) Relative expression of VDACs mRNA quantified by real-time PCR; (**B**) a, expression of VDAC isoforms and hexokinase II (HKII) in mitochondria-enriched fraction; b, co-immunoprecipitation of HK II and GSK-3β kinases with VDAC1; (**C**) A luminescent ATP detection kit was used to determine intracellular ATP levels in cells treated with pHaloTaq (control) or pmOGT-HaloTaq for 48 h. Data shown are averaged from at least three independent experiments performed in triplicate. * indicates significance, * *p* < 0.05; ** *p* < 0.01; *** *p* <0.001.

**Table 1 cancers-16-01048-t001:** The gold list of mOGT-modified ETC substrates. (The mapped O-GlcNAcylation and phosphorylation sites were obtained from the Human O-GlcNAcome Database ver. 1.3.).

L.p.	Complex type	Accession	Gene Symbol	Description	Coverage	MW [kDa]	O-GlcNAc sites	Phosphorylation Sites
1.	COMPLEX I	Q16795	NDUFA9	NADH dehydrogenase [ubiquinone] 1 alpha subcomplex subunit 9	37	42.5	-	-
2.	COMPLEX I	O95831	AIFM1	Apoptosis-inducing factor 1	40	66.9	S60	T105; S116; S118; S268; S292; S371; T521; S524; S530
3.	COMPLEX I	Q9H845	ACAD9	Complex I assembly factor ACAD9	77	68.7	S268; S302; S309	T478
4.	COMPLEX I	P03886	MT-ND1	NADH-ubiquinone oxidoreductase	49	35.6	-	-
5.	COMPLEX I	O95169	NDUFB8	NADH dehydrogenase [ubiquinone] 1 beta subcomplex subunit 8	51	21.8	T28	-
6.	COMPLEX I	O75306	NDUFS2	NADH dehydrogenase [ubiquinone] iron-sulfur protein 2	21	52.5	-	-
7.	COMPLEX I	O75489	NDUFS3	NADH dehydrogenase [ubiquinone] iron-sulfur protein 3	52	30.2	T21	-
8.	COMPLEX I	P03905	MT-ND4	NADH-ubiquinone oxidoreductase	46	51.5	-	-
9.	COMPLEX I	P03897	MT-ND3	NADH-ubiquinone oxidoreductase	87	13.2	T35	-
10.	COMPLEX I	P28331	NDUFS1	NADH-ubiquinone oxidoreductase 75 kDa subunit	11	79.4	S259; S318	-
11.	COMPLEX II	Q9NX18	SDHAF2	Succinate dehydrogenase assembly factor 2	59	19.6	-	-
12.	COMPLEX III	Q9BRT2	UQCC2	Ubiquinol-cytochrome-c reductase complex assembly factor 2	42	14.9	-	-
13.	COMPLEX III	P22695	UQCRC2	Cytochrome b-c1 complex subunit 2, mitochondrial	61	48.4	S367; T369	-
14.	COMPLEX IV	Q9Y276	BCS1L	Mitochondrial chaperone BCS1	42	47.5	-	T181
15.	COMPLEX IV	Q9BSH4	TAC01	Translational activator of cytochrome c oxidase 1	51	32.5	S2	-
16.	COMPLEX IV	Q00483	NDUFA4	Cytochrome c oxidase subunit NDUFA4	100	9.4	-	S66
17.	COMPLEX IV	Q96BR5	COA7	Cytochrome c oxidase assembly	100	25.7	-	-
18.	COMPLEX V	P06576	ATP5F1B	ATP synthase subunit beta,	16	56.5	S327	T312; S415; S433; S529
19.	COMPLEX V	Q5TC12	ATPAF1	ATP synthase mitochondrial F1 complex assembly factor 1	96	36.4	-	-

## Data Availability

The data presented in this study are available in this article (and Appendix A).

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
