# Peer review of "Increased O-GlcNAcylation by Upregulation of Mitochondrial O-GlcNAc Transferase (mOGT) Inhibits the Activity of Respiratory Chain Complexes and Controls Cellular Bioenergetics"

_cancers, 2024, doi:10.3390/cancers16051048_

Round 1

Reviewer 1 Report

Comments and Suggestions for Authors

The paper by Jozwiak et al describes how over-expression of mOGT affects mitochondrial function.  Although the paper is interesting, there are issues that dampen enthusiasm.  The primary concern is the issue of mOGT expression in general.  The authors site the Trapannone paper but fail to address the fact that human cells don't express mOGT.  The authors need to express a tagged version of mOGT because they can’t measure endogenous mOGT since it is not expressed.  By introducing a protein that is not normally found in cells, the authors could be introducing artifacts into their study.  Surprisingly, mOGT increased Complex 1 activity which is incongruent with other findings in the field.  

Other critiques: 

Figure 2 – the authors need to show the blots for O-GlcNAc, ncOGT, OGA, and phosphorylation from the purified mitochondria.  Further, the densitometry plots need to show the actual dot/value of the measurement.

Figure 4- the authors need to blot what they immune captured.  We need to know that the amount of protein pulled down was equivalent in each lane.  The O-GlcNAc and phosphorylation changes could be due to differences in the amounts pulled down.

Also the authors need to stress that they are not site mapping ETC chain protein O-GlcNAcylation.  Their method is indirect.

Figure 5 - The authors need to quantify the Ip blots in 5B-b

Author Response

  • The paper by Jozwiak et al describes how over-expression of mOGT affects mitochondrial function.  Although the paper is interesting, there are issues that dampen enthusiasm.  The primary concern is the issue of mOGT expression in general.  The authors site the Trapannone paper but fail to address the fact that human cells don't express mOGT.  The authors need to express a tagged version of mOGT because they can’t measure endogenous mOGT since it is not expressed.  By introducing a protein that is not normally found in cells, the authors could be introducing artifacts into their study.  

We agree with the reviewer that it would be pointless to study the function of mOGT in cells in which it does not occur naturally.  That's why, in the first step of our research, we checked whether this isoform is expressed in breast cells. The results confirming the endogenous expression of mOGT in breast cells were published in the article "Mitochondrial O-GlcNAc Transferase Interacts with and Modifies Many Proteins and Its Up-Regulation Affects Mitochondrial Function and Cellular Energy Homeostasis" - Jóźwiak P. et al, Cancers, 2021. In Figure 1, we have shown the endogenous level of mOGT, OGA and O-GlcNAc in mitochondria of breast cells. We have also visualized the transcript levels of mOGT and ncOGT using the RT-qPCR method.

  • Surprisingly, mOGT increased Complex 1 activity which is incongruent with other findings in the field.  

It appears that the effect of increased O-GlcNAcylation of proteins on complex I activity may vary depending on the cell type. In the Discussion, we have cited studies showing that increased O-GlcNAcylation in cardiomyocytes as a result of OGA inhibition leads to an increase in complex I activity. In contrast, transformed OGA knockout (KO) mouse embryonic fibroblasts (MEFs), which have a globally hyper-O-GlcNAcylated cellular environment, showed decreased complex I activity. We hypothesize that in breast cells with increased mOGT expression, the increase in complex I activity is related to the increase in mitochondrial transmembrane potential and mitochondrial ROS levels that we observed previously (Figure 3. in our previous article).

  • Figure 2 – the authors need to show the blots for O-GlcNAc, ncOGT, OGA, and phosphorylation from the purified mitochondria.  Further, the densitometry plots need to show the actual dot/value of the measurement.

Figure 2 demonstrates the impact of increased mOGT expression on the level of O-GlcNAcylation and phosphorylation of mitochondrial proteins. It is important to note that the O-GlcNAcylation level of mitochondrial proteins is not solely dependent on mOGT, but also on ncOGT and OGA. In previous work, we found that changes in endogenous mOGT expression did not affect the level of OGA expression in breast cancer mitochondria (see Figure 1C). Furthermore, we observed that treatment of cells with pmOGT-HaloTag had no effect on the level of endogenous OGT in breast cell mitochondria as well as ncOGT silencing using siRNA had no significant impact on O-GlcNAcylation levels in the mitochondrial fraction (see Figure 5). This information has been included in the Results section (lines 487 - 493).

  • Figure 4- the authors need to blot what they immune captured.  We need to know that the amount of protein pulled down was equivalent in each lane.  The O-GlcNAc and phosphorylation changes could be due to differences in the amounts pulled down.

The amount of immunocaptured respiratory chain proteins was compared to the breast cell lysates from which they were isolated, as shown in Figure 4 A and B. The Ponceau S-stained membranes were included below to confirm that the amount of proteins used for the immunoprecipitation was equal  in each line.

  • Also the authors need to stress that they are not site mapping ETC chain protein O-GlcNAcylation.  Their method is indirect.

Our study initially involved mapping O-GlcNAcylation sites of mitochondrial proteins. Unfortunately, the Click-iT™ O-GlcNAc Peptide LC/MS Standard (C33374) (Invitrogen)  which was used for  O-GlNAc sites mapping has been discontinued, thus we could not perform full analysis. We have updated the title of Table 1 to indicate that the mapped O-GlcNAcylation and phosphorylation sites were obtained from the Human O-GlcNAcome Database ver. 1.3.

  • Figure 5 - The authors need to quantify the Ip blots in 5B-b

Has been done. The Supplement includes densitometric analysis of the IP blots.

Reviewer 2 Report

Comments and Suggestions for Authors

The manuscript entitled " Increased O-GlcNAcylation by upregulation of mitochondrial O-GlcNAc transferase (mOGT) inhibits the activity of respiratory chain complexes and controls cellular bioenergetics" by Jóźwiak et. al highlights the role of mOGT in breast cancer cells and its role as a potential therapeutic target. The paper is very well written with one of the best introductions I have read so dar. They do a very good job in explaining most of their experiments and results. I have a few comments that I would like the authors to address before giving the green light for publications.

1. In results line 430 and 431, the authors mention the halo tag is on the N terminus but the graphical figure shows the tag on the C terminus.

2. In Figure1, panel E- is it possible to get better representative images. It is very difficult to see the cells from which the signal is coming from as well as the merged signal. Please update these images.

3. Figure 2, the entire figure lacks error bars and statistical data. It makes it difficult to interpret the data without these details. 

4. In the results, the authors mention that they used full length mOGT as well as the catalytically inactive mOGT. I'm surprised that the catalytically inactive mOGT also has an effect in decreasing activity of mitochondrial protein even though it should be inactive. Can the author provide justification as to why this may be?

5. In figure 5, the authors talk about the different VDACs expressed in cells and show their expression changes in context to mGOT over expression. They mention that VDAC3 is lost when mGOT is over expressed in all the three cell lines but they don't try to highlight the biological relevance of this and just move over to VDAC1 binding to hexokinase II and how it could impair ATP synthesis. Can the authors hightlight the relevance of loss of VDAC3, it seems to be a more dramatic effect.

Author Response

  1. In results line 430 and 431, the authors mention the halo tag is on the N terminus but the graphical figure shows the tag on the C terminus.

Has been corrected.

  1. In Figure1, panel E- is it possible to get better representative images. It is very difficult to see the cells from which the signal is coming from as well as the merged signal. Please update these images.

We have prepared new images presenting the results from confocal microscopy, as suggested.

  1. Figure 2, the entire figure lacks error bars and statistical data. It makes it difficult to interpret the data without these details. 

To isolate mitochondria from cells treated with plasmid DNA, we needed 40 x 106 cells. Therefore, we had to combine samples from several independent experiments. Consequently, these results cannot be statistically analyzed. However, our previous work titled "Mitochondrial O-GlcNAc Transferase Interacts with and Modifies Many Proteins and Its Up-Regulation Affects Mitochondrial Function and Cellular Energy Homeostasis", Jóźwiak P. et al, Cancers, 2021 (Figure 2), also shows the effect of mOGT on O-GlcNAcylation of mitochondrial proteins. In addition, the effect of mOGT on protein phosphorylation levels was assessed using two different methods: antibody and the Pro-Q Diamond reagent.

  1. In the results, the authors mention that they used full length mOGT as well as the catalytically inactive mOGT. I'm surprised that the catalytically inactive mOGT also has an effect in decreasing activity of mitochondrial protein even though it should be inactive. Can the author provide justification as to why this may be?

The primary function of OGT is to catalyze the attachment of the O-GlcNAc residue to proteins. However, recent evidence suggests that OGT may also have non-catalytic functions, and some discrepancies have been observed between genetic knockout of OGT and inhibition of its catalytic activity. While mouse embryonic fibroblasts (MEFs) die upon knockout, they do not when OGT is inhibited [Kazemi Z et al., 2010]. A recent study by Levine Z et al. (2020) showed that catalytically inactive OGT can cooperate within the same cellular pathway to alter cell physiology and is sufficient to drive MEF cell proliferation. They found that the non-catalytic functions of OGT are related to oxidative phosphorylation and components of the actin cytoskeleton. To exclude the non-catalytic effect of mOGT on respiratory chain activity, we used constructs that allow the expression of both mOGT and its non-catalytic mutant. This clarification has been added to the text in Results section (lines 434 - 444).

  1. In figure 5, the authors talk about the different VDACs expressed in cells and show their expression changes in context to mGOT over expression. They mention that VDAC3 is lost when mGOT is over expressed in all the three cell lines but they don't try to highlight the biological relevance of this and just move over to VDAC1 binding to hexokinase II and how it could impair ATP synthesis. Can the authors hightlight the relevance of loss of VDAC3, it seems to be a more dramatic effect.

We have added information on VDAC3 as recommended: lines 638 - 641 and lines 769 – 774 in Results and Discussion sections, respectively.

Round 2

Reviewer 1 Report

Comments and Suggestions for Authors

The authors address some the issues raised in the first report.  However, they state that the endogenous mOGT is blotted for in figure 1.  I can't find that.  I only see Halo-Tag OGT.  I still have strong reservations about mOGT being expressed in these cells.

Reviewer 2 Report

Comments and Suggestions for Authors

No comments, everything looks fine and all my concerns have been addressed.

Author Response

We appreciate the reviewer's valuable feedback, which has enhanced our manuscript.

Round 3

Reviewer 1 Report

Comments and Suggestions for Authors

Here is the problem, the only group in the field that shows the mito-OGT version is this group.  The van Aaltan group published a very good paper showing that the mito version off OGT should not be expressed in humans.  My group has never seen it.  I can’t rule out that some breast cancer cells lines can express it.  Anything is possible, but I have some concerns.  I think the group is looking at an artifact. However, the data does show a smaller oGT mito version, consistent with their hypothesis. What doesn’t make sense is why there is no full length OGT in the mitochondria.  Full-length OGT can be found in the mitochondria.  They don’t see it.  Why?  It’s hard to argue with their data since nothing is done experimentally wrong.  I’ll accept it.